# Convergence guarantees for kernel-based quadrature rules in misspecified settings

**Motonobu Kanagawa**[*], **Bharath K Sriperumbudur**[†], **Kenji Fukumizu**[*]
[*]The Institute of Statistical Mathematics, Tokyo 190-8562, Japan
[†]Department of Statistics, Pennsylvania State University, University Park, PA 16802, USA
kanagawa@ism.ac.jp, bks18@psu.edu, fukumizu@ism.ac.jp

## Abstract

Kernel-based quadrature rules are becoming important in machine learning and statistics, as they achieve super-$\sqrt{n}$ convergence rates in numerical integration, and thus provide alternatives to Monte Carlo integration in challenging settings where integrands are expensive to evaluate or where integrands are high dimensional. These rules are based on the assumption that the integrand has a certain degree of smoothness, which is expressed as that the integrand belongs to a certain reproducing kernel Hilbert space (RKHS). However, this assumption can be violated in practice (e.g., when the integrand is a black box function), and no general theory has been established for the convergence of kernel quadratures in such misspecified settings. Our contribution is in proving that kernel quadratures can be consistent even when the integrand does not belong to the assumed RKHS, i.e., when the integrand is less smooth than assumed. Specifically, we derive convergence rates that depend on the (unknown) lesser smoothness of the integrand, where the degree of smoothness is expressed via powers of RKHSs or via Sobolev spaces.

## 1 Introduction

Numerical integration, or quadrature, is a fundamental task in the construction of various statistical and machine learning algorithms. For instance, in Bayesian learning, numerical integration is generally required for the computation of marginal likelihood in model selection, and for the marginalization of parameters in fully Bayesian prediction, etc [20]. It also offers flexibility to probabilistic modeling, since, e.g., it enables us to use a prior that is not conjugate with a likelihood function.

Let $P$ be a (known) probability distribution on a measurable space $\mathcal{X}$ and $f$ be an integrand on $\mathcal{X}$. Suppose that the integral $\int f(x)dP(x)$ has no closed form solution. One standard form of numerical integration is to approximate the integral as a weighted sum of function values $f(X_1), \ldots, f(X_n)$ by appropriately choosing the points $X_1, \ldots, X_n \in \mathcal{X}$ and weights $w_1, \ldots, w_n \in \mathbb{R}$:

$$\sum_{i=1}^{n} w_i f(X_i) \approx \int f(x)dP(x). \qquad (1)$$

For example, the simplest Monte Carlo method generates the points $X_1, \ldots, X_n$ as an i.i.d. sample from $P$ and uses equal weights $w_1 = \cdots = w_n = 1/n$. Convergence rates of such Monte Carlo methods are of the form $n^{-1/2}$, which can be slow for practical purposes. For instance, in situations where the evaluation of the integrand requires heavy computations, $n$ should be small and Monte Carlo would perform poorly; such situations typically appear in modern scientific and engineering applications, and thus quadratures with faster convergence rates are desirable [18].

One way of achieving faster rates is to exploit one's prior knowledge or assumption about the integrand (e.g. the degree of smoothness) in the construction of a weighted point set $\{(w_i, X_i)\}_{i=1}^{n}$.

Reproducing kernel Hilbert spaces (RKHS) have been successfully used for this purpose, with examples being Quasi Monte Carlo (QMC) methods based on RKHSs [13] and Bayesian quadratures [19]; see e.g. [11, 6] and references therein. We will refer to such methods as *kernel-based quadrature rules* or simply *kernel quadratures* in this paper. A kernel quadrature assumes that the integrand $f$ belongs to an RKHS consisting of smooth functions (such as Sobolev spaces), and constructs the weighted points $\{(w_i, X_i)\}_{i=1}^n$ so that the *worst case error* in that RKHS is small. Then the error rate of the form $n^{-b}$, $b \geq 1$, which is much faster than the rates of Monte Carlo methods, will be guaranteed with $b$ being a constant representing the degree of smoothness of the RKHS (e.g., the order of differentiability). Because of this nice property, kernel quadratures have been studied extensively in recent years [7, 3, 5, 2, 17] and have started to find applications in machine learning and statistics [23, 14, 12, 6].

However, if the integrand $f$ does *not* belong to the assumed RKHS (i.e. if $f$ is less smooth than assumed), there is no known theoretical guarantee for fast convergence rate or even the consistency of kernel quadratures. Such misspecification is likely to happen if one does not have the full knowledge of the integrand—such situations typically occur when the integrand is a *black box function*. As an illustrative example, let us consider the problem of illumination integration in computer graphics (see e.g. Sec. 5.2.4 of [6]). The task is to compute the total amount of light arriving at a camera in a virtual environment. This is solved by numerical integration with integrand $f(x)$ being the intensity of light arriving at the camera from a direction $x$ (angle). The value $f(x)$ is only given by simulation of the environment for each $x$, so $f$ is a black box function. In such a situation, one's assumption on the integrand can be misspecified. Establishing convergence guarantees for such misspecified settings has been recognized as an important open problem in the literature [6, Section 6].

**Contributions.** The main contribution of this paper is in providing general convergence guarantees for kernel-based quadrature rules in misspecified settings. Specifically, we make the following contributions:

- In Section 4, we prove that consistency can be guaranteed even when the integrand $f$ does not belong to the assumed RKHS. Specifically, we derive a convergence rate of the form $n^{-\theta b}$, where $0 < \theta \leq 1$ is a constant characterizing the (relative) smoothness of the integrand. In other words, the integration error decays at a speed depending on the (unknown) smoothness of the integrand. This guarantee is applicable to kernel quadratures that employ *random* points.

- We apply this result to QMC methods called *lattice rules* (with randomized points) and the quadrature rule by Bach [2], for the setting where the RKHS is a Korobov space. We show that even when the integrand is less smooth than assumed, the error rate becomes the same as for the case when the (unknown) smoothness is known; namely, we show that these methods are *adaptive* to the unknown smoothness of the integrand.

- In Section 5, we provide guarantees for kernel quadratures with *deterministic* points, by focusing on specific cases where the RKHS is a Sobolev space $W_2^r$ of order $r \in \mathbb{N}$ (the order of differentiability). We prove that consistency can be guaranteed even if $f \in W_2^s \backslash W_2^r$ where $s \leq r$, i.e., the integrand $f$ belongs to a Sobolev space $W_2^s$ of lesser smoothness. We derive a convergence rate of the form $n^{-bs/r}$, where the ratio $s/r$ determines the relative degree of smoothness.

- As an important consequence, we show that if weighted points $\{(w_i, X_i)\}_{i=1}^m$ achieve the optimal rate in $W_2^r$, then they also achieve the optimal rate in $W_2^s$. In other words, to achieve the optimal rate for the integrand $f$ belonging to $W_2^s$, one does *not* need to know the smoothness $s$ of the integrand; one only needs to know its upper-bound $s \leq r$.

This paper is organized as follows. In Section 2, we describe kernel-based quadrature rules, and formally state the goal and setting of theoretical analysis in Section 3. We present our contributions in Sections 4 and 5. Proofs are collected in the supplementary material.

**Related work.** Our work is close to [17] in spirit, which discusses situations where the true integrand is *smoother* than assumed (which is complementary to ours) and proposes a control functional approach to make kernel quadratures adaptive to the (unknown) greater smoothness. We also note that there are certain quadratures which are adaptive to less smooth integrands [8, 9, 10]. On the other hand, our aim here is to provide general theoretical guarantees that are applicable to a wide class of kernel-based quadrature rules.

**Notation.** For $x \in \mathbb{R}^d$, let $\{x\} \in [0,1]^d$ be its fractional parts. For a probability distribution $P$ on a measurable space $\mathcal{X}$, let $L_2(P)$ be the Hilbert space of square-integrable functions with respect to $P$. If $P$ is the Lebesgue measure on $\mathcal{X} \subset \mathbb{R}^d$, let $L_2(\mathcal{X}) := L_2(P)$ and further $L_2 := L_2(\mathbb{R}^d)$. Let $C_0^s := C_0^s(\mathbb{R}^d)$ be the set of all functions on $\mathbb{R}^d$ that are continuously differentiable up to order $s \in \mathbb{N}$ and vanish at infinity. Given a function $f$ and weighted points $\{(w_i, X_i)\}_{i=1}^n$, $Pf := \int f(x)dP(x)$ and $P_n f := \sum_{i=1}^n w_i f(X_i)$ denote the integral and its numerical approximation, respectively.

## 2 Kernel-based quadrature rules

Suppose that one has prior knowledge on certain properties of the integrand $f$ (e.g. its order of differentiability). A kernel quadrature exploits this knowledge by expressing it as that $f$ belongs to a certain RKHS $\mathcal{H}$ that possesses those properties, and then constructing weighted points $\{(w_i, X_i)\}_{i=1}^n$ so that the error of integration is small for every function in the RKHS. More precisely, it pursues the minimax strategy that aims to minimize the *worst case error* defined by

$$e_n(P; \mathcal{H}) := \sup_{f \in \mathcal{H}: \|f\|_{\mathcal{H}} \leq 1} |Pf - P_n f| := \sup_{f \in \mathcal{H}: \|f\|_{\mathcal{H}} \leq 1} \left| \int f(x)dP(x) - \sum_{i=1}^n w_i f(X_i) \right|, \quad (2)$$

where $\| \cdot \|_{\mathcal{H}}$ denotes the norm of $\mathcal{H}$. The use of RKHS is beneficial (compared to other function spaces), because it results in an analytic expression of the worst case error (2) in terms of the reproducing kernel. Namely, one can explicitly compute (2) in the construction of $\{(w_i, X_i)\}_{i=1}^n$ as a criterion to be minimized. Below we describe this as well as examples of kernel quadratures.

### 2.1 The worst case error in RKHS

Let $\mathcal{X}$ be a measurable space and $\mathcal{H}$ be an RKHS of functions on $\mathcal{X}$ with $k : \mathcal{X} \times \mathcal{X} \to \mathbb{R}$ as the bounded reproducing kernel. By the reproducing property, it is easy to verify that $Pf = \langle f, m_P \rangle_{\mathcal{H}}$ and $P_n f = \langle f, m_{P_n} \rangle_{\mathcal{H}}$ for all $f \in \mathcal{H}$, where $\langle \cdot, \cdot \rangle_{\mathcal{H}}$ denotes the inner-product of $\mathcal{H}$, and

$$m_P := \int k(\cdot, x)dP(x) \in \mathcal{H}, \quad m_{P_n} := \sum_{i=1}^n w_i k(\cdot, X_i) \in \mathcal{H}.$$

Therefore, the worst case error (2) can be written as the difference between $m_P$ and $m_{P_n}$:

$$\sup_{\|f\|_{\mathcal{H}} \leq 1} |Pf - P_n f| = \sup_{\|f\|_{\mathcal{H}} \leq 1} \langle f, m_P - m_{P_n} \rangle_{\mathcal{H}} = \|m_P - m_{P_n}\|_{\mathcal{H}}, \quad (3)$$

where $\| \cdot \|_{\mathcal{H}}$ denotes the norm of $\mathcal{H}$ defined by $\|f\|_{\mathcal{H}} = \sqrt{\langle f, f \rangle_{\mathcal{H}}}$ for $f \in \mathcal{H}$. From (3), it is easy to see that for every $f \in \mathcal{H}$, the integration error $|P_n f - Pf|$ is bounded by the worst case error:

$$|P_n f - Pf| \leq \|f\|_{\mathcal{H}} \|m_P - m_{P_n}\|_{\mathcal{H}} = \|f\|_{\mathcal{H}} e_n(P; \mathcal{H}).$$

We refer the reader to [21, 11] for details of these derivations. Using the reproducing property of $k$, the r.h.s. of (3) can be alternately written as:

$$e_n(P; \mathcal{H}) = \sqrt{\int \int k(x, \tilde{x})dP(x)dP(\tilde{x}) - 2 \sum_{i=1}^n w_i \int k(x, X_i)dP(x) + \sum_{i=1}^n \sum_{j=1}^n w_i w_j k(X_i, X_j)}.$$
$$(4)$$

The integrals in (4) are known in closed form for many pairs of $k$ and $P$; see e.g. Table 1 of [6]. For instance, if $P$ is the uniform distribution on $\mathcal{X} = [0,1]^d$, and $k$ is the Korobov kernel described below, then $\int k(y, x)dP(x) = 1$ for all $y \in \mathcal{X}$. To pursue the aforementioned minimax strategy, one can explicitly use the formula (4) to minimize the worst case error (2). Often $\mathcal{H}$ is chosen as an RKHS consisting of smooth functions, and the degree of smoothness is what a user specifies; we describe this in the example below.

### 2.2 Examples of RKHSs: Korobov spaces

The setting $\mathcal{X} = [0,1]^d$ is standard in the literature on numerical integration; see e.g. [11]. In this setting, *Korobov spaces* and *Sobolev spaces* have been widely used as RKHSs.[1] We describe the former here; for the latter, see Section 5.

**Korobov space on** $[0, 1]$. The Korobov space $W_{\text{Kor}}^{\alpha}([0, 1])$ of order $\alpha \in \mathbb{N}$ is an RKHS whose kernel $k_{\alpha}$ is given by

$$k_{\alpha}(x, y) := 1 + \frac{(-1)^{\alpha-1}(2\pi)^{2\alpha}}{(2\alpha)!} B_{2\alpha}(\{x - y\}), \tag{5}$$

where $B_{2\alpha}$ denotes the $2\alpha$-th Bernoulli polynomial. $W_{\text{Kor}}^{\alpha}([0, 1])$ consists of periodic functions on $[0, 1]$ whose derivatives up to $(\alpha - 1)$-th are absolutely continuous and the $\alpha$-th derivative belongs to $L_2([0, 1])$ [16]. Therefore the order $\alpha$ represents the degree of smoothness of functions in $W_{\text{Kor}}^{\alpha}([0, 1])$.

**Korobov space on** $[0, 1]^d$. For $d \geq 2$, the kernel of the Korobov space is given as the product of one-dimensional kernels (5):

$$k_{\alpha,d}(x, y) := k_{\alpha}(x_1, y_1) \cdots k_{\alpha}(x_d, y_d), \ x := (x_1, \ldots, x_d)^T, \ y := (y_1, \ldots, y_d)^T \in [0, 1]^d. \tag{6}$$

The induced Korobov space $W_{\text{Kor}}^{\alpha}([0, 1]^d)$ on $[0, 1]^d$ is then the tensor product of one-dimensional Korobov spaces: $W_{\text{Kor}}^{\alpha}([0, 1]^d) := W_{\text{Kor}}^{\alpha}([0, 1]) \otimes \cdots \otimes W_{\text{Kor}}^{\alpha}([0, 1])$. Therefore it consists of functions having square-integrable mixed partial derivatives up to the order $\alpha$ in *each* variable. This means that by using the kernel (6) in the computation of (4), one can make an assumption that the integrand $f$ has smoothness of degree $\alpha$ in each variable. In other words, one can incorporate one's knowledge or belief on $f$ into the construction of weighted points $\{(w_i, X_i)\}$ via the choice of $\alpha$.

### 2.3 Examples of kernel-based quadrature rules

We briefly describe examples of kernel-based quadrature rules.

**Quasi Monte Carlo (QMC).** These methods typically focus on the setting where $\mathcal{X} = [0, 1]^d$ with $P$ being the uniform distribution on $[0, 1]^d$, and employ equal weights $w_i = \cdots = w_n = 1/n$. Popular examples are *lattice rules* and *digital nets/sequences*. Points $X_1, \ldots, X_n$ are selected in a deterministic way so that the worst case error (4) is as small as possible. Then such deterministic points are often *randomized* to obtain unbiased integral estimators, as we will explain in Section 4.2. For a review of these methods, see [11].

For instance, lattice rules generate $X_1, \ldots, X_n$ in the following way (for simplicity assume $n$ is prime). Let $z \in \{1, \ldots, n-1\}^d$ be a generator vector. Then the points are defined as $X_i = \{iz/n\} \in [0, 1]^d$ for $i = 1, \ldots, n$. Here $z$ is selected so that the resulting worst case error (2) becomes as small as possible. The CBC (Component-By-Component) construction is a fast method that makes use of the formula (4) to achieve this; see Section 5 of [11] and references therein. Lattice rules applied to the Korobov space $W_{\text{Kor}}^{\alpha}([0, 1]^d)$ can achieve the rate $e_n(P, W_{\text{Kor}}^{\alpha}([0, 1]^d) = O(n^{-\alpha+\xi})$ for the worst case error with $\xi > 0$ arbitrarily small [11, Theorem 5.12].

**Bayesian quadratures.** These methods are applicable to general $\mathcal{X}$ and $P$, and employ non-uniform weights. Points $X_1, \ldots, X_n$ are selected either deterministically or randomly. Given the points being fixed, weights $w_1, \ldots, w_n$ are obtained by minimizing (4), which can be done by solving a linear system of size $n$. Such methods are called Bayesian quadratures, since the resulting estimate $P_n f$ in this case is exactly the posterior mean of the integral $Pf$ given "observations" $\{(X_i, f(X_i))\}_{i=1}^n$, with the prior on the integrand $f$ being Gaussian Process with the covariance kernel $k$. We refer to [6] for these methods.

For instance, the algorithm by Bach [2] proceeds as follows, for the case of $\mathcal{H}$ being a Korobov space $W_{\text{Kor}}^{\alpha}([0, 1]^d)$ and $P$ being the uniform distribution on $[0, 1]^d$: (i) Generate points $X_1, \ldots, X_n$ independently from the uniform distribution on $[0, 1]^d$; (ii) Compute weights $w_1, \ldots, w_n$ by minimizing (4), with the constraint $\sum_{i=1}^n w_i^2 \leq 4/n$. Bach [2] proved that this procedure gives the error rate $e_n(P, W_{\text{Kor}}^{\alpha}([0, 1]^d) = O(n^{-\alpha+\xi})$ for $\xi > 0$ arbitrarily small.[2]

## 3 Setting and objective of theoretical analysis

We now formally state the setting and objective of our theoretical analysis in general form. Let $P$ be a known distribution and $\mathcal{H}$ be an RKHS. Our starting point is that weighted points $\{(w_i, X_i)\}_{i=1}^n$

are already constructed for each $n \in \mathbb{N}$ by some quadrature rule[3], and that these provide consistent approximation of $P$ in terms of the worst case error:

$$e_n(P; \mathcal{H}) = \|m_P - m_{P_n}\|_{\mathcal{H}} = O(n^{-b}) \quad (n \to \infty), \tag{7}$$

where $b > 0$ is some constant. Here we do not specify the quadrature algorithm explicitly, to establish results applicable to a wide class of kernel quadratures simultaneously.

Let $f$ be an integrand that is *not* included in the RKHS: $f \notin \mathcal{H}$. Namely, we consider a *misspecified setting*. Our aim is to derive convergence rates for the integration error

$$|P_n f - P f| = \left| \sum_{i=1}^{n} w_i f(X_i) - \int f(x) dP(x) \right|$$

based on the assumption (7). This will be done by assuming a certain regularity condition on $f$, which expresses (unknown) lesser smoothness of $f$. For example, this is the case when the weighted points are constructed by assuming the Korobov space of order $\alpha \in \mathbb{N}$, but the integrand $f$ belongs to the Korobov space of order $\beta < \alpha$: in this case, $f$ is less smooth than assumed. As mentioned in Section 1, such misspecification is likely to happen if $f$ is a black box function. But misspecification also occurs even when one has the full knowledge of $f$. As explained in Section 2.1, the kernel $k$ should be chosen so that the integrals in (4) allow analytic solutions w.r.t. $P$. Namely, the distribution $P$ determines an available class of kernels (e.g. Gaussian kernels for a Gaussian distribution), and therefore the RKHS of a kernel from this class may not contain the integrand of interest. This situation can be seen in application to random Fourier features [23], for example.

## 4 Analysis 1: General RKHS with random points

We first focus on kernel quadratures with random points. To this end, we need to introduce certain assumptions on (i) the construction of weighted points $\{(w_i, X_i)\}_{i=1}^{n}$ and on (ii) the smoothness of the integrand $f$; we discuss them in Sections 4.1 and 4.2, respectively. In particular, we introduce the notion of *powers of RKHSs* [22] in Section 4.2, which enables us to characterize the (relative) smoothness of the integrand. We then state our main result in Section 4.3, and illustrate it with QMC lattice rules (with randomization) and the Bayesian quadrature by Bach [2] in Korobov RKHSs.

### 4.1 Assumption on random points $X_1, \ldots, X_n$

**Assumption 1.** *There exists a probability distribution $Q$ on $\mathcal{X}$ satisfying the following properties: (i) $P$ has a bounded density function w.r.t. $Q$; (ii) there is a constant $D > 0$ independent of $n$, such that*

$$\left( \mathbb{E} \left[ \frac{1}{n} \sum_{i=1}^{n} g^2(X_i) \right] \right)^{1/2} \leq D\|g\|_{L_2(Q)}, \quad \forall g \in L_2(Q), \tag{8}$$

*where $\mathbb{E}[\cdot]$ denotes the expectation w.r.t. the joint distribution of $X_1, \ldots, X_n$.*

Assumption 1 is fairly general, as it does not specify any distribution of points $X_1, \ldots, X_n$, but just requires that the expectations over these points satisfy (8) for some distribution $Q$ (also note that it allows the points to be dependent). For instance, let us consider the case where $X_1, \ldots, X_n$ are independently generated from a user-specified distribution $Q$; in this case, $Q$ serves as a proposal distribution. Then (8) holds for $D = 1$ with equality. Examples in this case include the Bayesian quadratures by Bach [2] and Briol et al. [6] with random points.

Assumption 1 is also satisfied by QMC methods that apply randomization to deterministic points, which is common in the literature [11, Sections 2.9 and 2.10]. Popular methods for randomization are *random shift* and *scrambling*, both of which satisfy Assumption 1 for $D = 1$ with equality, where $Q (= P)$ is the uniform distribution on $\mathcal{X} = [0,1]^d$. This is because in general, randomization is applied to make an integral estimator unbiased: $\mathbb{E}[\frac{1}{n} \sum_{i=1}^{n} f(X_i)] = \int_{[0,1]^d} f(x) dx$ [11, Section 2.9]. For instance, the random shift is done as follows. Let $x_1, \ldots, x_n \in [0,1]^d$ be deterministic points generated by a QMC method. Let $\Delta$ be a random sample from the uniform distribution on $[0,1]^d$. Then each $X_i$ is given as $X_i := \{x_i + \Delta\} \in [0,1]^d$. Therefore $\mathbb{E}[\frac{1}{n} \sum_{i=1}^{n} g^2(X_i)] = \int_{[0,1]^d} g^2(x) dx = \int g^2(x) dQ(x)$ for all $g \in L_2(Q)$, so (8) holds for $D = 1$ with equality.

## 4.2 Assumption on the integrand via powers of RKHSs

To state our assumption on the integrand $f$, we need to introduce *powers of RKHSs* [22, Section 4]. Let $0 < \theta \leq 1$ be a constant. First, with the distribution $Q$ in Assumption 1, we require that the kernel satisfies

$$\int k(x, x) dQ(x) < \infty.$$

For example, this is always satisfied if the kernel is bounded. We also assume that the support of $Q$ is entire $\mathcal{X}$ and that $k$ is continuous. These conditions imply *Mercer's theorem* [22, Theorem 3.1 and Lemma 2.3], which guarantees the following expansion of the kernel $k$:

$$k(x, y) = \sum_{i=1}^{\infty} \mu_i e_i(x) e_i(y), \quad x, y \in \mathcal{X}, \tag{9}$$

where $\mu_1 \geq \mu_2 \geq \cdots > 0$ and $\{e_i\}_{i=1}^{\infty}$ is an orthonormal series in $L_2(Q)$; in particular, $\{\mu_i^{1/2} e_i\}_{i=1}^{\infty}$ forms an orthonormal basis of $\mathcal{H}$. Here the convergence of the series in (9) is pointwise. Assume that $\sum_{i=1}^{\infty} \mu_i^{\theta} e_i^2(x) < \infty$ holds for all $x \in \mathcal{X}$. Then *the $\theta$-th power of the kernel $k$* is a function $k^{\theta} : \mathcal{X} \times \mathcal{X} \to \mathbb{R}$ defined by

$$k^{\theta}(x, y) := \sum_{i=1}^{\infty} \mu_i^{\theta} e_i(x) e_i(y), \quad x, y \in \mathcal{X}. \tag{10}$$

This is again a reproducing kernel [22, Proposition 4.2], and defines an RKHS called *the $\theta$-th power of the RKHS $\mathcal{H}$*:

$$\mathcal{H}^{\theta} = \left\{ \sum_{i=1}^{\infty} a_i \mu_i^{\theta/2} e_i : \sum_{i=1}^{\infty} a_i^2 < \infty \right\}.$$

This is an intermediate space between $L_2(Q)$ and $\mathcal{H}$, and the constant $0 < \theta \leq 1$ determines how close $\mathcal{H}^{\theta}$ is to $\mathcal{H}$. For instance, if $\theta = 1$ we have $\mathcal{H}^{\theta} = \mathcal{H}$, and $\mathcal{H}^{\theta}$ approaches $L_2(Q)$ as $\theta \to +0$. Indeed, $\mathcal{H}^{\theta}$ is nesting w.r.t. $\theta$:

$$\mathcal{H} = \mathcal{H}^1 \subset \mathcal{H}^{\theta} \subset \mathcal{H}^{\theta'} \subset L_2(Q), \quad \text{for all } 0 < \theta' < \theta < 1. \tag{11}$$

In other words, $\mathcal{H}^{\theta}$ gets larger as $\theta$ decreases. If $\mathcal{H}$ is an RKHS consisting of smooth functions, then $\mathcal{H}^{\theta}$ contains less smooth functions than those in $\mathcal{H}$; we will show this in the example below.

**Assumption 2.** *The integrand $f$ lies in $\mathcal{H}^{\theta}$ for some $0 < \theta \leq 1$.*

We note that Assumption 2 is equivalent to assuming that $f$ belongs to the *interpolation space* $[L_2(Q), \mathcal{H}]_{\theta, 2}$, or lies in the range of a power of certain integral operator [22, Theorem 4.6].

**Powers of Tensor RKHSs.** Let us mention the important case where RKHS $\mathcal{H}$ is given as the tensor product of individual RKHSs $\mathcal{H}_1, \ldots, \mathcal{H}_d$ on the spaces $\mathcal{X}_1, \ldots, \mathcal{X}_d$, i.e., $\mathcal{H} = \mathcal{H}_1 \otimes \cdots \otimes \mathcal{H}_d$ and $\mathcal{X} = \mathcal{X}_1 \times \cdots \times \mathcal{X}_d$. In this case, if the distribution $Q$ is the product of individual distributions $Q_1, \ldots, Q_d$ on $\mathcal{X}_1, \ldots, X_n$, it can be easily shown that the power RKHS $\mathcal{H}^{\theta}$ is the tensor product of individual power RKHSs $\mathcal{H}_i^{\theta}$:

$$\mathcal{H}^{\theta} = \mathcal{H}_1^{\theta} \otimes \cdots \otimes \mathcal{H}_d^{\theta}. \tag{12}$$

**Examples: Powers of Korobov spaces.** Let us consider the Korobov space $W_{\mathrm{Kor}}^{\alpha}([0,1]^d)$ with $Q$ being the uniform distribution on $[0,1]^d$. The Korobov kernel (5) has a Mercer representation

$$k_{\alpha}(x, y) = 1 + \sum_{i=1}^{\infty} \frac{1}{i^{2\alpha}} [c_i(x) c_i(y) + s_i(x) s_i(y)], \tag{13}$$

where $c_i(x) := \sqrt{2} \cos 2\pi i x$ and $s_i(x) := \sqrt{2} \sin 2\pi i x$. Note that $c_0(x) := 1$ and $\{c_i, s_i\}_{i=1}^{\infty}$ constitute an orthonormal basis of $L_2([0,1])$. From (10) and (13), the $\theta$-th power of the Korobov kernel $k_{\alpha}$ is given by

$$k_{\alpha}^{\theta}(x, y) = 1 + \sum_{i=1}^{\infty} \frac{1}{i^{2\alpha\theta}} [c_i(x) c_i(y) + s_i(x) s_i(y)].$$

If $\alpha\theta \in \mathbb{N}$, this is exactly the kernel $k_{\alpha\theta}$ of the Korobov space $W_{\mathrm{Kor}}^{\alpha\theta}([0,1])$ of lower order $\alpha\theta$. In other words, $W_{\mathrm{Kor}}^{\alpha\theta}([0,1])$ is nothing but the $\theta$-power of $W_{\mathrm{Kor}}^{\alpha}([0,1])$. From this and (12), we can also show that the $\theta$-th power of $W_{\mathrm{Kor}}^{\alpha}([0,1]^d)$ is $W_{\mathrm{Kor}}^{\alpha\theta}([0,1]^d)$ for $d \geq 2$.

### 4.3 Result: Convergence rates for general RKHSs with random points

The following result guarantees the consistency of kernel quadratures for integrands satisfying Assumption 2, i.e., $f \in \mathcal{H}^\theta$.

**Theorem 1.** *Let $\{(w_i, X_i)\}_{i=1}^n$ be such that $\mathbb{E}[e_n(P; \mathcal{H})] = O(n^{-b})$ for some $b > 0$ and $\mathbb{E}[\sum_{i=1}^n w_i^2] = O(n^{-2c})$ for some $0 < c \leq 1/2$, as $n \to \infty$. Assume also that $\{X_i\}_{i=1}^n$ satisfies Assumption 1. Let $0 < \theta \leq 1$ be a constant. Then for any $f \in \mathcal{H}^\theta$, we have*

$$\mathbb{E}\left[|P_n f - P f|\right] = O(n^{-\theta b + (1/2-c)(1-\theta)}) \quad (n \to \infty). \tag{14}$$

**Remark 1.** *(a) The expectation in the assumption $\mathbb{E}[e_n(P; \mathcal{H})] = O(n^{-b})$ is w.r.t. the joint distribution of the weighted points $\{(w_i, X_i)\}_{i=1}^n$.*

*(b) The assumption $\mathbb{E}[\sum_{i=1}^n w_i^2] = O(n^{-2c})$ requires that each weight $w_i$ decreases as $n$ increases. For instance, if $\max_{i \in \{1,...,n\}} |w_i| = O(n^{-1})$, we have $c = 1/2$. For QMC methods, weights are uniform $w_i = 1/n$, so we always have $c = 1/2$. The quadrature rule by Bach [2] also satisfies $c = 1/2$; see Section 2.3.*

*(c) Let $c = 1/2$. Then the rate in (14) becomes $O(n^{-\theta b})$, which shows that the integral estimator $P_n f$ is consistent, even when the integrand $f$ does not belong to $\mathcal{H}$ (recall $\mathcal{H} \subsetneq \mathcal{H}^\theta$ for $\theta < 1$; see also (11)). The resulting rate $O(n^{-\theta b})$ is determined by $0 < \theta \leq 1$ of the assumption $f \in \mathcal{H}^\theta$, which characterizes the closeness of $f$ to $\mathcal{H}$.*

*(d) When $\theta = 1$ (well-specified case), irrespective of the value of $c$, the rate in (14) becomes $O(n^{-b})$, which recovers the rate of the worst case error $\mathbb{E}[e_n(P; \mathcal{H})] = O(n^{-b})$.*

**Examples in Korobov spaces.** Let us illustrate Theorem 1 in the following setting described earlier. Let $\mathcal{X} = [0, 1]^d$, $\mathcal{H} = W_{\text{Kor}}^\alpha([0, 1]^d)$, and $P$ be the uniform distribution on $[0, 1]^d$. Then $\mathcal{H}^\theta = W_{\text{Kor}}^{\alpha\theta}([0, 1]^d)$, as discussed in Section 4.2. Let us consider the two methods discussed in Section 2.3: (i) the QMC lattice rules with randomization and (ii) the algorithm by Bach [2]. For both the methods, we have $c = 1/2$, and the distribution $Q$ in Assumption 1 is uniform on $[0, 1]^d$ in this setting. As mentioned before, these methods achieve the rate $n^{-\alpha+\xi}$ for arbitrarily small $\xi > 0$ in the well-specified setting: $b = \alpha - \xi$ in our notation.

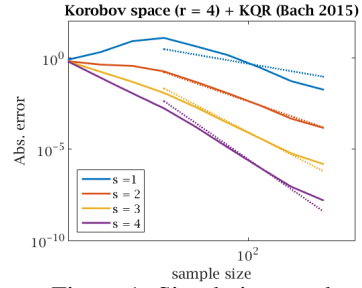

Figure 1: Simulation results

Then the assumption $f \in \mathcal{H}^\theta$ reads $f \in W_{\text{Kor}}^{\alpha\theta}([0, 1]^d)$ for $0 < \theta \leq 1$. For such an integrand $f$, we obtain the rate $O(n^{-\alpha\theta+\xi})$ in (14) with arbitrarily small $\xi > 0$. This is the *same* rate as for a well-specified case where $W_2^{\alpha\theta}([0, 1]^d)$ was assumed for the construction of weighted points. Namely, we have shown that these methods are adaptive to the unknown smoothness of the integrand.

For the algorithm by Bach [2], we conducted simulation experiments to support this observation, by using code available from `http://www.di.ens.fr/~fbach/quadrature.html`. The setting is what we have described with $d = 1$, and weights are obtained without regularization as in [2]. The result is shown in Figure 1, where $r (= \alpha)$ denotes the assumed smoothness, and $s (= \alpha\theta)$ is the (unknown) smoothness of an integrand. The straight lines are (asymptotic) upper-bounds in Theorem 1 (slope $-s$ and intercept fitted for $n \geq 2^4$), and the corresponding solid lines are numerical results (both in log-log scales). Averages over 100 runs are shown. The result indeed shows the adaptability of the quadrature rule by Bach for the less smooth functions (i.e. $s = 1, 2, 3$). We observed similar results for the QMC lattice rules (reported in Appendix D in the supplement).

## 5 Analysis 2: Sobolev RKHS with deterministic points

In Section 4, we have provided guarantees for methods that employ random points. However, the result does not apply to those with *deterministic* points, such as (a) QMC methods without randomization, (b) Bayesian quadratures with deterministic points, and (c) kernel herding [7].

We aim here to provide guarantees for quadrature rules with deterministic points. To this end, we focus on the setting where $\mathcal{X} = \mathbb{R}^d$ and $\mathcal{H}$ is a *Sobolev space* [1]. The Sobolev space $W_2^r$ of order $r \in \mathbb{N}$ is defined by

$$W_2^r := \{f \in L_2 : D^\alpha f \in L_2 \text{ exists for all } |\alpha| \leq r\}$$

where $\alpha := (\alpha_1, \ldots, \alpha_d)$ with $\alpha_i \geq 0$ is a multi-index with $|\alpha| := \sum_{i=1}^{d} \alpha_i$, and $D^\alpha f$ is the $\alpha$-th (weak) derivative of $f$. Its norm is defined by $\|f\|_{W_2^r} = (\sum_{|\alpha| \leq r} \|D^\alpha f\|_{L_2}^2)^{1/2}$. For $r > d/2$, this is an RKHS with the reproducing kernel $k$ being the *Matèrn kernel*; see Section 4.2.1. of [20] for the definition.

Our assumption on the integrand $f$ is that it belongs to a Sobolev space $W_2^s$ of a lower order $s \leq r$. Note that the order $s$ represents the smoothness of $f$ (the order of differentiability). Therefore the situation $s < r$ means that $f$ is *less* smooth than assumed; we consider the setting where $W_2^r$ was assumed for the construction of weighted points.

**Rates under an assumption on weights.** The first result in this section is based on the same assumption on weights as in Theorem 1.

**Theorem 2.** *Let $\{(w_i, X_i)\}_{i=1}^n$ be such that $e_n(P; W_2^r) = O(n^{-b})$ for some $b > 0$ and $\sum_{i=1}^{n} w_i^2 = O(n^{-2c})$ for some $0 < c \leq 1/2$, as $n \to \infty$. Then for any $f \in C_0^s \cap W_2^s$ with $s \leq r$, we have*

$$|P_n f - Pf| = O(n^{-bs/r + (1/2 - c)(1 - s/r)}) \quad (n \to \infty). \tag{15}$$

**Remark 2.** *(a) Let $\theta := s/r$. Then the rate in (15) is rewritten as $O(n^{-\theta b + (1/2 - c)(1 - \theta)})$, which matches the rate of Theorem 1. In other words, Theorem 2 provides a deterministic version of Theorem 1 for the special case of Sobolev spaces.*

*(b) Theorem 2 can be applied to quadrature rules with equally-weighted deterministic points, such as QMC methods and kernel herding [7]. For these methods, we have $c = 1/2$ and so we obtain the rate $O(n^{-sb/r})$ in (15). The minimax optimal rate in this setting (i.e., $c = 1/2$) is given by $n^{-b}$ with $b = r/d$ [15]. For these choices of $b$ and $c$, we obtain a rate of $O(n^{-s/d})$ in (15), which is exactly the optimal rate in $W_2^s$. This leads to an important consequence that the optimal rate $O(n^{-s/d})$ can be achieved for an integrand $f \in W_2^s$ without knowing the degree of smoothness $s$; one just needs to know its upper-bound $s \leq r$. Namely, any methods of optimal rates in Sobolev spaces are adaptive to lesser smoothness.*

**Rates under an assumption on separation radius.** Theorems 1 and 2 require the assumption $\sum_{i=1}^{n} w_i^2 = O(n^{-2c})$. However, for some algorithms, the value of $c$ may not be available. For instance, this is the case for Bayesian quadratures that compute the weights without any constraints [6]; see Section 2.3. Here we present a preliminary result that does not rely on the assumption on the weights. To this end, we introduce a quantity called *separation radius*:

$$q_n := \min_{i \neq j} \|X_i - X_j\|.$$

In the result below, we assume that $q_n$ does not decrease very quickly as $n$ increases. Let $\mathrm{diam}(X_1, \ldots, X_n)$ denote the diameter of the points.

**Theorem 3.** *Let $\{(w_i, X_i)\}_{i=1}^n$ be such that $e_n(P; W_2^r) = O(n^{-b})$ for some $b > 0$ as $n \to \infty$, $q_n \geq Cn^{-b/r}$ for some $C > 0$, and $\mathrm{diam}(X_1, \ldots, X_n) \leq 1$. Then for any $f \in C_0^s \cap W_2^s$ with $s \leq r$, we have*

$$|P_n f - Pf| = O(n^{-\frac{bs}{r}}) \quad (n \to \infty). \tag{16}$$

Consequences similar to those of Theorems 1 and 2 can be drawn for Theorem 3. In particular, the rate in (16) coincides with that of (15) with $c = 1/2$. The assumption $q_n \geq Cn^{-b/r}$ can be verified when points form equally-spaced grids in a compact subset of $\mathbb{R}^d$. In this case, the separation radius satisfies $q_n \geq Cn^{-1/d}$ for some $C > 0$. As above, the optimal rate for this setting is $n^{-b}$ with $b = r/d$, which implies the separation radius satisfies the assumption as $n^{-b/r} = n^{-1/d}$.

## 6 Conclusions

Kernel quadratures are powerful tools for numerical integration. However, their convergence guarantees had not been established in situations where integrands are less smooth than assumed, which can happen in various situations in practice. In this paper, we have provided the first known theoretical guarantees for kernel quadratures in such misspecified settings.

**Acknowledgments**

We wish to thank the anonymous reviewers for valuable comments. We also thank Chris Oates for fruitful discussions. This work has been supported in part by MEXT Grant-in-Aid for Scientific Research on Innovative Areas (25120012).

## Footnotes

[1] Korobov spaces are also known as *periodic Sobolev spaces* in the literature [4, p.318].

[2]Note that in [2], the degree of smoothness is expressed in terms of $s := \alpha d$.

[3]Note that here the weighted points should be written as $\{(w_i^{(n)}, X_i^{(n)})\}_{i=1}^{n}$, since they are constructed depending on the number of points $n$. However, we just write as $\{(w_i, X_i)\}_{i=1}^{n}$ for notational simplicity.

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
