[Supplementary Material · supplement_final.pdf]

# Supplementary materials for "Convergence guarantees for kernel-based quadrature rules in misspecified settings"

**Motonobu Kanagawa**$^*$, **Bharath K Sriperumbudur**$^\dagger$, **Kenji Fukumizu**$^*$
$^*$The Institute of Statistical Mathematics, Tokyo 190-8562, Japan
$^\dagger$Department of Statistics, Pennsylvania State University, University Park, PA 16802, USA
kanagawa@ism.ac.jp, bks18@psu.edu, fukumizu@ism.ac.jp

**Notation.** Below we use the notation $L_1 := L_1(\mathbb{R}^d)$ and $L_2 := L_2(\mathbb{R}^d)$. For a function $f \in L_1$, we denote by $\hat{f}$ its Fourier transform. $B(0,R) := \{x \in \mathbb{R}^d : \|x\| \le R\}$ denotes the ball of radius $R > 0$ centered at $0$. $\mathrm{supp}(f)$ denotes the support of a function $f$.

## A  Proof of Theorem 1

*Proof.* First we define the following integral operator $T : L_2(Q) \to L_2(Q)$:

$$Tf := \int k(\cdot, x) f(x) dQ(x), \quad f \in L_2(Q).$$

From Lemmas 2.2 and 2.3 of Steinwart and Scovel (2012), the condition $\int k(x,x)dQ(x) < \infty$ guarantees that $T$ is compact, positive, and self-adjoint. Therefore $T$ allows an eigen decomposition $T = \sum_{i=1}^{\infty} \mu_i \langle e_i, \cdot \rangle_{L_2(Q)} e_i$, where $\mu_1 \ge \mu_2, \cdots \ge 0$ are eigenvalues of $T$ and $e_i$ is an eigenfunction associated with $\mu_i$. Based on this decomposition, define an operator $T^{\frac{\theta}{2}} : L_2(Q) \to L_2(Q)$ by $T^{\frac{\theta}{2}} := \sum_{i=1}^{\infty} \mu_i^{\frac{\theta}{2}} \langle e_i, \cdot \rangle_{L_2(Q)} e_i$. Let $\mathcal{R}(T^{\frac{\theta}{2}})$ denote the range of $T^{\frac{\theta}{2}}$. From Lemma 6.4 of Steinwart and Scovel (2012), we have $\mathcal{R}(T^{\frac{\theta}{2}}) = \mathcal{H}^{\theta}$. Therefore the assumption $f \in \mathcal{H}^{\theta}$ is equivalent to $f \in \mathcal{R}(T^{\frac{\theta}{2}})$.

For each $n$, define a constant $\lambda_n = n^{-2b+2c-1}$. Define a function $f_{\lambda_n} \in \mathcal{H}$ by $f_{\lambda_n} := (T + \lambda_n I)^{-1} T f$, where $I$ denotes the identity. From Lemma 3 of Smale and Zhou (2007) (see also Theorem 4 and Eq. (7.10) of Smale and Zhou (2005)) and $f \in \mathcal{R}(T^{\frac{\theta}{2}})$, $f_{\lambda_n}$ satisfies

$$\|f - f_{\lambda_n}\|_{L_2(Q)} \le \lambda_n^{\frac{\theta}{2}} \|T^{-\frac{\theta}{2}} f\|_{L_2(Q)}, \tag{1}$$

$$\|f_{\lambda_n}\|_{\mathcal{H}} \le \lambda_n^{-\frac{1-\theta}{2}} \|T^{-\frac{\theta}{2}} f\|_{L_2(Q)}. \tag{2}$$

It follows from triangle inequality that

$$\mathbb{E}[|P_n f - Pf|] \le \underbrace{\mathbb{E}[|P_n f - P_n f_{\lambda_n}|]}_{(A)} + \underbrace{\mathbb{E}[|P_n f_{\lambda_n} - P f_{\lambda_n}|]}_{(B)} + \underbrace{\mathbb{E}[|P f_{\lambda_n} - Pf|]}_{(C)}. \tag{3}$$

Below we separately bound terms $(A)$, $(B)$ and $(C)$.

$$
\begin{aligned}
(A) \;&=\; \mathbb{E}\left[\left|\left|\sum_{i=1}^{n} w_i(f(X_i) - f_{\lambda_n}(X_i))\right|\right|\right] \\
&\leq\; \mathbb{E}\left[\left(\sum_{i=1}^{n} w_i^2\right)^{1/2}\left(\sum_{i=1}^{n}(f(X_i) - f_{\lambda_n}(X_i))^2\right)^{1/2}\right] \quad (\because \text{Cauchy}-\text{Schwartz}) \\
&\leq\; \left(\mathbb{E}\left[\sum_{i=1}^{n} w_i^2\right]\right)^{1/2}\left(\mathbb{E}\left[n\left(\frac{1}{n}\sum_{i=1}^{n}(f(X_i) - f_{\lambda_n}(X_i))^2\right)\right]\right)^{1/2} \quad (\because \text{Cauchy}-\text{Schwartz}) \\
&\leq\; \left(\mathbb{E}\left[\sum_{i=1}^{n} w_i^2\right]\right)^{1/2} n^{1/2} D \|f - f_{\lambda_n}\|_{L_2(Q)} \quad (\because \text{Assumption 1}) \\
&=\; O(n^{-c+1/2}\lambda_n^{\theta/2}) \quad (\because (1)) \\
&=\; O(n^{-\theta b+(1/2-c)(1-\theta)}).
\end{aligned}
$$

We now bound $(B)$ as

$$
\begin{aligned}
(B) \;&=\; \mathbb{E}\left[\langle m_{P_n} - m_P, f_{\lambda_n}\rangle_{\mathcal{H}}\right] \quad (\because f_{\lambda_n} \in \mathcal{H}) \\
&\leq\; \mathbb{E}\left[\|m_{P_n} - m_P\|_{\mathcal{H}}\right]\|f_{\lambda_n}\|_{\mathcal{H}} \\
&=\; O(n^{-b}\lambda_n^{-\frac{1-\theta}{2}}) \quad (\because (2)) \\
&=\; O(n^{-\theta b+(1/2-c)(1-\theta)}).
\end{aligned}
$$

To bound $(C)$, let $r$ denote the (bounded) density function of $P$ with respect to $Q$: $dP(x) = r(x)dQ(x)$.

$$
\begin{aligned}
(C) \;&\leq\; \|f_{\lambda_n} - f\|_{L_1(P)} \leq \|f_{\lambda_n} - f\|_{L_2(P)} \\
&=\; \left(\int (f_{\lambda_n}(x) - f(x))^2 r(x)dQ(x)\right)^{1/2} \\
&\leq\; \|r\|_{L_\infty}\|f_{\lambda_n} - f\|_{L_2(Q)} = O(\lambda_n^{\theta/2}) \quad (\because (1)).
\end{aligned}
$$

Note that $(C)$ decays faster than $(A)$ since $c \leq 1/2$, and so the rate is dominated by $(A)$ and $(B)$. The proof is completed by substituting these terms in (3). $\qquad\square$

## B  Proof of Theorem 2

### B.1  Approximation in Sobolev spaces

In the proof of Theorem 2, we will use Proposition 3.7 of Narcowich and Ward (2004), which assumes the existence of a function $\psi : \mathbb{R}^d \to \mathbb{R}$ satisfying the properties in Lemma 1. Since the existence of this function is not proved in Narcowich and Ward (2004), we will first prove it for completeness. Lemma 1 is a variant of Lemma (1.1) of Frazier et al. (1991), from which we borrowed the proof idea.

**Lemma 1.** *Let $s$ be any positive integer. Then there exists a function $\psi : \mathbb{R}^d \to \mathbb{R}$ satisfying the following properties:*
*(a) $\psi$ is radial;*
*(b) $\psi$ is a Schwartz function;*
*(c) $\mathrm{supp}(\hat\psi) \subset B(0,1)$;*
*(d) $\int_{\mathbb{R}^d} x^\beta \psi(x)dx = 0$ for every multi index $\beta$ satisfying $|\beta| := \sum_{i=1}^{d}\beta_i \leq s$.*
*(e) $\psi$ satisfies*

$$
\int_0^\infty |\hat\psi(t\xi)|^2 \frac{dt}{t} = 1, \quad \forall \xi \in \mathbb{R}^d\backslash\{0\}. \tag{4}
$$

*Proof.* Let $u : \mathbb{R}^d \to \mathbb{R}$ be the function whose Fourier transform is given by $\hat{u}(\xi) := \exp(-\frac{1}{1-|\xi|})$ for $|\xi| < 1$ and $\hat{u}(\xi) = 0$ for $|\xi| \geq 1$. Then $\hat{u}$ is radial, Schwartz, and satisfies $\mathrm{supp}(\hat{u}) \subset B(0,1)$. Also note that since $\hat{u}$ is symmetric $u$ is real-valued.

Define a function $h : \mathbb{R}^d \to \mathbb{R}$ by $\Delta^m u$ for some $m \in \mathbb{N}$ satisfying $m > s/2$, where $\Delta$ denotes the Laplace operator. From $\hat{h}(\xi) = (-1)^m |\xi|^{2m} \hat{u}(\xi)$ (e.g. p. 117 of Stein (1970)), it is immediate that $\hat{h}$ is radial and Schwartz (and so is $h$), and that $\mathrm{supp}(\hat{h}) \subset B(0,1)$. Thus $h$ satisfies (a) (b) and (c).

We next show that $h$ satisfies (d) (which can instead be shown using the integration by parts). Using the notation $p_\beta(x) := x^\beta$, we have $\int_{\mathbb{R}} x^\beta h(x) dx = (2\pi)^{d/2} \widehat{p_\beta h}(0)$ since $h$ is Schwartz (see e.g. Theorem 5.20 and p.75 of Wendland (2005)). Since $\widehat{p_\beta h}(\xi) = i^{|\beta|} D_\beta \hat{h}(\xi)$ (where $D_\beta$ denotes the mixed partial derivatives with multi index $\beta$; see e.g. Theorem 5.16 of Wendland (2005)) and $\hat{h}(\xi) = (-1)^m |\xi|^{2m} \hat{u}$ with $2m > s \geq |\beta|$ and $\hat{u}$ being bounded, $\widehat{p_\beta h}(\xi)$ is bounded by a polynomial of order $2m - |\beta| > 0$ without a degree 0 term. Therefore $\widehat{p_\beta h}(0) = 0$, which implies (d).

Next, we show that $\int_0^\infty |\hat{h}(t\xi)|^2 \frac{dt}{t} < \infty$ for all $\xi \in \mathbb{R}^d \backslash \{0\}$. Since $\hat{h}$ is bounded and $\mathrm{supp}(\hat{h}) \subset B(0,1)$, we have $\int_1^\infty |\hat{h}(t\xi)|^2 \frac{dt}{t} < \infty$. Also, since $|\hat{h}(t\xi)| = O(t^{2m})$ as $t \to +0$ (which follows from $\hat{h}(t\xi) = (-1)^m |t\xi|^{2m} \hat{u}(t\xi)$ with $\hat{u}$ being bounded), we have $\int_0^1 |\hat{h}(t\xi)|^2 \frac{dt}{t} < \infty$. Therefore $\int_0^\infty |\hat{h}(t\xi)|^2 \frac{dt}{t} < \infty$.

Note that since $\hat{h}$ is radial, $\int_0^\infty |\hat{h}(t\xi)|^2 \frac{dt}{t}$ only depends on the norm $|\xi|$. Furthermore, $\int_0^\infty |\hat{h}(t\xi)|^2 \frac{dt}{t}$ remains the same for different values of the norm $|\xi| > 0$ due to the property of the Haar measure $dt/t$. In other words, there is a constant $0 < C < \infty$ satisfying $\int_0^\infty |\hat{h}(t\xi)|^2 \frac{dt}{t} = C$ for all $\xi \in \mathbb{R}^d \backslash \{0\}$. The proof completes by defining $\psi$ in the assertion by $\psi(x) := C^{-1/2} h(x)$. $\square$

Note that $\psi$ being radial implies that $\hat{\psi}$ is radial, so $\hat{\psi}(t\xi)$ in (4) depends on $\xi$ only through its norm $|\xi|$. Therefore henceforth we will use the notation $\hat{\psi}(t|\xi|)$ to denote $\hat{\psi}(t\xi)$, to emphasize its dependence on the norm. Similarly, we use the notation $\hat{\psi}(t)$ to imply $\hat{\psi}(t\xi)$ for some $\xi \in \mathbb{R}^d$ with $|\xi| = 1$.

**Approximation via Calderón's formula.** If $\psi \in L_1$ is radial and satisfies (4), Calderón's formula (Frazier et al., 1991, Theorem 1.2) guarantees that any $f \in L_2$ can be written as

$$f(x) = \int_0^\infty (\psi_t * \psi_t * f)(x) \, \frac{dt}{t},$$

where $\psi_t(x) := \frac{1}{t^d} \psi(x/t)$. This equality should be interpreted in the following $L_2$ sense: if $0 < \varepsilon < \delta < \infty$ and $f_{\varepsilon,\delta}(x) := \int_\varepsilon^\delta (\psi_t * \psi_t * f)(x) \frac{dt}{t}$, then $\|f - f_{\varepsilon,\delta}\|_{L_2} \to 0$ as $\varepsilon \to 0$ and $\delta \to \infty$.

Following Section 3.2 of Narcowich and Ward (2004), we now take $\psi$ from Lemma 1 and consider the following approximation of $f$:

$$g_\sigma(x) := \int_{1/\sigma}^\infty (\psi_t * \psi_t * f)(x) \, \frac{dt}{t}. \tag{5}$$

We will need the following lemma (which is not given in Narcowich and Ward (2004)).

**Lemma 2.** *Let $0 < s \leq r$ and $\sigma > 0$ be constants. If $f \in W_2^s$, the function $g_\sigma$ defined in (5) satisfies*

$$\|g_\sigma\|_{W_2^r} \leq (1 + \sigma^2)^{\frac{r-s}{2}} \|f\|_{W_2^s}.$$

*Proof.* The Fourier transform of $g_\sigma$ can be written as

$$\hat{g}_\sigma(\xi) = \hat{f}(\xi) \int_{1/\sigma}^\infty |\hat{\psi}(t|\xi|)|^2 \frac{dt}{t} = \hat{f}(\xi) \begin{cases} 0 & \text{if } |\xi| \geq \sigma, \\ \int_{|\xi|/\sigma}^1 |\hat{\psi}(t)|^2 \frac{dt}{t} & \text{if } |\xi| < \sigma. \end{cases}$$

In other words, $\text{supp}(\hat{g}_\sigma) \subset B(0, \sigma)$. Also note that for $|\xi| < \sigma$, we have $\int_{|\xi|/\sigma}^{1} |\hat{\psi}(t)|^2 \frac{dt}{t} \leq \int_0^1 |\hat{\psi}(t)|^2 \frac{dt}{t} \leq 1$ from (4). Therefore,

$$
\begin{aligned}
\|g_\sigma\|_{W_2^r}^2 &= \int_0^\sigma (1 + |\xi|^2)^r |\hat{g_\sigma}(\xi)|^2 d\xi \\
&\leq \int_0^\sigma (1 + |\xi|^2)^r |\hat{f}(\xi)|^2 d\xi \\
&= \int_0^\sigma (1 + |\xi|^2)^{r-s} (1 + |\xi|^2)^s |\hat{f}(\xi)|^2 d\xi \\
&\leq (1 + \sigma^2)^{r-s} \int_0^\sigma (1 + |\xi|^2)^s |\hat{f}(\xi)|^2 d\xi \\
&\leq (1 + \sigma^2)^{r-s} \|f\|_{W_2^s}^2.
\end{aligned}
$$

$\square$

## B.2  Proof of Theorem 2

We are now ready to prove Theorem 2.

*Proof.* For each $n$, define a constant $\sigma_n = n^{\frac{b-c+1/2}{r}}$. Let $g_{\sigma_n} \in W_2^r$ be an approximation of $f$ defined in (5) with $\sigma = \sigma_n$. Then from Proposition 3.7 of Narcowich and Ward (2004), it satisfies

$$\|f - g_{\sigma_n}\|_{L_\infty} \leq C\sigma_n^{-s} \|f\|_{C_0^s}, \tag{6}$$

where $C$ is a constant depending only on $s$ and $f$. From Lemma 2 in Appendix B.1, $g_{\sigma_n}$ also satisfies

$$\|g_{\sigma_n}\|_{W_2^r} \leq (1 + \sigma_n^2)^{\frac{r-s}{2}} \|f\|_{W_2^s}. \tag{7}$$

By triangle inequality, we have

$$|P_n f - Pf| \leq \underbrace{|P_n f - P_n g_{\sigma_n}|}_{(A)} + \underbrace{|P_n g_{\sigma_n} - P g_{\sigma_n}|}_{(B)} + \underbrace{|P g_{\sigma_n} - Pf|}_{(C)}. \tag{8}$$

Below we separately bound terms $(A)$, $(B)$ and $(C)$.

$$
\begin{aligned}
(A) &= \sum_{i=1}^n w_i(f(X_i) - g_{\sigma_n}(X_i)) \leq \sum_{i=1}^n |w_i| \|f - g_{\sigma_n}\|_{L_\infty} \\
&\leq n^{1/2} (\sum_{i=1}^n w_i^2)^{1/2} \|f - g_{\sigma_n}\|_{L_\infty} = O(n^{1/2-c}\sigma_n^{-s}) \quad (\because (6)) \\
&= O(n^{-bs/r + (1/2-c)(1-s/r)}).
\end{aligned}
$$

$$
\begin{aligned}
(B) &= \langle g_{\sigma_n}, m_{P_n} - m_P \rangle_{W_2^r} \quad (\because g_{\sigma_n} \in W_2^r) \\
&\leq \|g_{\sigma_n}\|_{W_2^r} \|m_{P_n} - m_P\|_{W_2^r} = O(\sigma_n^{r-s} n^{-b}) \quad (\because (7)) \\
&= O(n^{-bs/r + (1/2-c)(1-s/r)}).
\end{aligned}
$$

$$
(C) = \int (g_{\sigma_n}(x) - f(x)) dP(x) \leq \|g_{\sigma_n} - f\|_{L_\infty} = O(\sigma_n^{-s}) \quad (\because (6)).
$$

Since $c \leq 1/2$, note that $(C)$ decays faster than $(A)$ and so the rate is dominated by $(A)$ and $(B)$. The proof is completed by inserting these terms in (8).

$\square$

# C  Proof of Theorem 3

*Proof.* Define a constant $C_d \sigma_n = n^{b/r}$, where $C_d := 24(\frac{\sqrt{\pi}}{3}\Gamma(\frac{d+2}{2}))^{\frac{2}{d+1}}$ with $\Gamma$ being the Gamma function, so that $\sigma_n \geq C_d/q_n$. Then from Theorem 3.5 and Theorem 3.10 of Narcowich and Ward (2004), there exists a function $f_{\sigma_n} \in W_2^r$ satisfying the following properties:

$$f_{\sigma_n}(X_i) = f(X_i), \quad (i = 1, \ldots, n), \tag{9}$$

$$\|f - f_{\sigma_n}\|_{L_\infty} \leq C_{s,d}\sigma_n^{-s}\max(\|f\|_{C_0^s}, \|f\|_{W_2^s}), \tag{10}$$

where $C_{s,d}$ is a constant only depending on $s$ and $d$.

Moreover, from the discussion in p. 298 of Narcowich and Ward (2004), this function also satisfies

$$\|f_{\sigma_n}\|_{W_2^r} \leq C\sigma_n^{r-s}\max(\|f\|_{C_0^s}, \|f\|_{W_2^s}), \tag{11}$$

where $C$ is a constant only depending on $s$, $d$, and the kernel $k$.

Triangle inequality yields

$$|P_n f - Pf| \leq \underbrace{|P_n f - P_n f_{\sigma_n}|}_{(A)} + \underbrace{|P_n f_{\sigma_n} - P f_{\sigma_n}|}_{(B)} + \underbrace{|P f_{\sigma_n} - Pf|}_{(C)}. \tag{12}$$

We separately bound terms $(A)$, $(B)$ and $(C)$.

$$(A) = \sum_{i=1}^{n} w_i(f(X_i) - f_{\sigma_n}(X_i)) = 0. \quad (\because (9)).$$

$$
\begin{aligned}
(B) &= \langle f_{\sigma_n}, m_{P_n} - m_P \rangle_{W_2^r} \quad (\because f_{\sigma_n} \in W_2^r) \\
&\leq \|f_{\sigma_n}\|_{W_2^r} \|m_{P_n} - m_P\|_{W_2^r} = O(\sigma_n^{r-s}n^{-b}) \quad (\because (11)) \\
&= O(n^{-bs/r}).
\end{aligned}
$$

$$
\begin{aligned}
(C) &= \int (f_{\sigma_n}(x) - f(x))dP(x) \leq \|f_{\sigma_n} - f\|_{L_\infty} \\
&\leq C_{s,d}\sigma_n^{-s}\max(\|f\|_{C_0^s}, \|f\|_{W_2^s}) \quad (\because (10)) \\
&= O(n^{-bs/r}).
\end{aligned}
$$

The proof is completed by inserting these bounds in (12)

$\square$

# D  Experimental results for QMC lattice rules

We conducted simulation experiments with QMC lattice rules to show their adaptability to integrands less smooth than assumed. The RKHS is the Korobov space of dimension $d = 2$. For the construction of generator vectors, we employed a fast method for component-by-component (CBC) construction by Nuyens (2007), using the code provided at `https://people.cs.kuleuven.be/~dirk.nuyens/fast-cbc/`. This method constructs a generator vector for lattice points whose number $n$ is prime. Here we did not apply randomization to the generated points, so they were deterministic. We would like to note that this situation is not covered by our current theoretical guarantees; the results in Section 4 only apply to random points, and those in Section 5 apply to deterministic points with Sobolev spaces.

The results are shown in Figure 1, where $r (= \alpha)$ denotes the assumed smoothness, and $s (= \alpha\theta)$ is the (unknown) smoothness of an integrand. The straight lines are (asymptotic) upper-bounds in Theorem 1 (slope $-s$ and intercept fitted for $n \geq 2^4$), and the corresponding solid lines are numerical results (both in log-log scales). For $s = 4$ with large sample sizes, underflow occurred with our computational environment as the errors were quite small, so we do not report these results. The results indicate the adaptability of the QMC lattice rules for the less smooth functions (i.e. $s = 1, 2, 3$).

Figure 1: Simulation results for QMC lattice rules by fast CBC construction (Nuyens, 2007)