[Reviews · NeurIPS 2016]

Reviewer 1

Summary

The paper studies convergence rates for kernel based quadrature-rules in misspecified settings. The problem consists in estimating int_X f(x) dP(x), with sum_{i=1}^n w_i f(x_i) for a suitable set x_1, ... x_n and weights w_1, ... w_n The setting is misspecified when the integrand f does not belong to a given RKHS. The proposed analysis covers 1) Monte Carlo (i.i.d. random points sampled according to P and w_i = 1/n) 2) QMC (quasi-random points and w_i = 1/n) 3) i.i.d. points w.r.t. P, with w_i selected with kernel ridge regression.

Qualitative Assessment

The paper analyzes theoretically different quadrature-rules studying the technical case where the integrand does not belong to a given RKHS used by the method. The proposed theorems achieve optimal convergence rate in the known subcases of the used setting. The proofs seem to be correct and the the presentation is rather clear. It is not very clear the practical impact of the technical case considered.

Confidence in this Review

3-Expert (read the paper in detail, know the area, quite certain of my opinion)


Reviewer 2

Summary

%%% UPDATE %%% Thank you for your concise response. I remain confident that this paper would make a solid contribution to NIPS this year. %%% END %%% This paper extends results on kernel quadrature to the misspecified setting, where the integrand is not smooth enough to belong to the RKHS H characterised by the kernel. The main result of the paper is that kernel quadrature is adaptive to unknown smoothness; this is provided that the integrand belongs to a RKHS on the Hilbert scale between H and L2.

Qualitative Assessment

I would like to congratulate the authors on an excellent paper. This is an important contribution to the literature on kernel quadrature and established useful theoretical guidance to practitioners - namely, to err on the side of using kernels that are `too smooth' and allow automatic adaptation to occur. Given the importance of numerical integration, I expect this work to have substantial impact. I have some very minor comments. The literature tends to refer to "kernel quadrature" when the weights w_i are those that minimise the worst-case error for fixed locations X_i, i.e. the weights w_i that are used in Bayesian quadrature. This agrees with your definition on page 3, line 105. However, defined in this way, quasi Monte Carlo (QMC) is *not* kernel quadrature (the weights are uniform weights). I do take the authors' point that QMC can be studied in RKHS, but I would clarify the wording on page 1, line 26 to emphasise that QMC is not itself a kernel quadrature method. This work is close in spirit to ref [17] in the paper. In [17] it was argued that a QMC lattice rule for functions that are rougher than the integrand could be compensated for by building control variates with a kernel that was smoother than the integrand. The argument of adapting to unknown smoothness was also made in [17]. I believe the present paper implies that an alternative to [17] is to simply use a QMC lattice rule that is for smoother functions. This work is perhaps a nicer solution compared to [17]. On page 3, line 102, the phrase "it suffices" is a bit misleading. You are *choosing* to pursue a minimax strategy, but until this point the goal was not stated in terms this explicit. In the supplement on page 1, line 13, I don't think Smale and Zhou (2007) is the original reference for this result (they cite another work).

Confidence in this Review

2-Confident (read it all; understood it all reasonably well)


Reviewer 3

Summary

This is an extension of [1]. In [1] kernel based quadrature rules are studied for the case that the integrand lies in the RKHS. This paper extends this result to the adaptive setting where the integrand does not necessarily lie in the RKHS but in a hierarchy of spaces given through the interpolation space [L2,H]_{theta,2}. Theta controls here the rate, i.e. the smaller theta the "further away" we are from H and the slower the rate of convergence. [1] On the Equivalence between Kernel Quadrature Rules and Random Feature Expansions, F. Bach.

Qualitative Assessment

The extension of [1] to the adaptive case is natural and interesting. In my opinion it is important that methods work with as few assumptions as possible and the adaptive setting is the right way to minimise assumptions. The use of interpolation spaces is also the standard approach to address adaptivity. I wasn't aware of the link between powers of RKHSs and interpolation spaces. The link is useful since it allows one to move from a particular interpolation space to a sequence of (rescaled) eigenvalues and corresponding eigenfunctions. I appreciate the technical part of the paper and I consider the paper useful. However, it is somewhat on the incremental side given [1]. Ideally, this work will be accepted as a poster presentation. [1] On the Equivalence between Kernel Quadrature Rules and Random Feature Expansions, F. Bach.

Confidence in this Review

2-Confident (read it all; understood it all reasonably well)


Reviewer 4

Summary

The authors focus on providing general convergence guarantees for kernel-based quadrature rules in misspecified settings. The goal is to approximate the integral $\int f(x) \, \mathrm d \mu(x)$ with a weighted sum of function values in the case where the integrand $f$ is not as smooth as assumed.

Qualitative Assessment

The paper is badly written. The problem is not clearly explained, especially from a statistical point of view, and in my opinion as it is described it is not a relevant problem. Terminology is not correct or it is not used in an appropriate way. As it is written, the paper missed a theoretical background from a statistical point of view. post-rebuttal: authors provide satisfying explanations. I have increased my score concerning "clarity", "potential impact" and "technical quality".

Confidence in this Review

2-Confident (read it all; understood it all reasonably well)


Reviewer 5

Summary

The paper increases the theoretical flexibility of "kernel-based quadrature rules", a class of numerical integration methods. It derives new results on the performance of these methods when traditional assumptions do not hold. Specifically, it presents a variety of guarantees relating integrand smoothness to convergence rates of these methods, where previous work instead simply assumed smoothness. The (claimed) widespread use of these methods in machine learning means the results should be interesting to a significant portion of the NIPS audience.

Qualitative Assessment

I'm an outsider to this area so, taking the paper's overview of the subject at its word, it appears to me that this work fills a meaningful gap in the area and provides useful theoretical guarantees where previously there were none. If the numerical integration methods discussed in the paper are indeed as widespread in machine learning as claimed, then I think the paper deserves the sort of machine learning audience NIPS provides. This is why I have leaned toward recommending this paper. However: since this is not my area, I am not confident that my interpretation above is correct. I don't really know how novel this type of argument is, and I don't really know how useful the results are. This is why I can only make a weak recommendation, and have given 3 scores for technical quality, novelty, and usefulness. Regarding this last point: as a non-expert I would have appreciated a bit more language about why these results are relevant to various areas of machine learning. In particular, while the paper mentions that smoothness "assumption[s] can be violated", I only see two sentences (lines 43-46) that tell me anything about when, and why one might care about that. For a general audience, I think a better explanation of why the results matter would help. This is why I have given a 2 score for clarity and presentation. EDIT (Post-rebuttal): After seeing the authors' rebuttal and promise to better contextualize this contribution to machine learning in general, I have increased my Clarity score from 2 to 3.

Confidence in this Review

1-Less confident (might not have understood significant parts)